# Minimal lactazole scaffold for in vitro thiopeptide bioengineering

Alexander A. Vinogradov [1,5], Morito Shimomura[2,5], Yuki Goto [1✉], Taro Ozaki[2,4], Shumpei Asamizu[2,3], Yoshinori Sugai[2], Hiroaki Suga [1✉] & Hiroyasu Onaka [2,3✉]

Lactazole A is a cryptic thiopeptide from *Streptomyces lactacystinaeus*, encoded by a compact 9.8 kb biosynthetic gene cluster. Here, we establish a platform for in vitro biosynthesis of lactazole A, referred to as the FIT-Laz system, via a combination of the flexible in vitro translation (FIT) system with recombinantly produced lactazole biosynthetic enzymes. Systematic dissection of lactazole biosynthesis reveals remarkable substrate tolerance of the biosynthetic enzymes and leads to the development of the minimal lactazole scaffold, a construct requiring only 6 post-translational modifications for macrocyclization. Efficient assembly of such minimal thiopeptides with FIT-Laz opens access to diverse lactazole analogs with 10 consecutive mutations, 14- to 62-membered macrocycles, and 18 amino acid-long tail regions, as well as to hybrid thiopeptides containing non-proteinogenic amino acids. This work suggests that the minimal lactazole scaffold is amenable to extensive bioengineering and opens possibilities to explore untapped chemical space of thiopeptides.

¹ Department of Chemistry, Graduate School of Science, The University of Tokyo, Bunkyo-ku, Tokyo 113-0033, Japan. ² Department of Biotechnology, Graduate School of Agricultural and Life Sciences, The University of Tokyo, Bunkyo-ku, Tokyo 113-8657, Japan. ³ Collaborative Research Institute for Innovative Microbiology, The University of Tokyo, Bunkyo-ku, Tokyo 113-8657, Japan. ⁴ Department of Chemistry, Graduate School of Science, Hokkaido University, Sapporo, Hokkaido 060-0810, Japan. ⁵These authors contributed equally: Alexander A. Vinogradov, Morito Shimomura. ✉email: y-goto@chem.s.u-tokyo.ac.jp; hsuga@chem.s.u-tokyo.ac.jp; aonaka@mail.ecc.u-tokyo.ac.jp

Thiopeptides are natural products defined by a six-membered nitrogenous heterocycle, usually pyridine, grafted within the backbone of a peptidic macrocycle[1]. Multiple azole rings, dehydroamino acids, and other optional non-proteinogenic elements further contribute to the resulting structural complexity characteristic of thiopeptides. More than a hundred thiopeptides isolated to date possess strong antibiotic activity against Gram-positive bacteria, including methicillin-resistant *Staphylococcus aureus* (MRSA)[1]. For instance, thiostrepton has been used as a topical antibiotic in veterinary medicine and LFF571, a synthetic derivative of naturally occurring GE2270A, underwent clinical trials as a treatment against *Clostridium difficile* infections[2].

A decade ago, thiopeptides were shown to be of ribosomal origin[3–6]. During biosynthesis, a structural gene encoding a thiopeptide precursor is transcribed and translated, and the resulting peptide undergoes posttranslational modifications (PTMs) introduced by cognate enzymes colocalized with the structural gene in a biosynthetic gene cluster (BGC). Commonly, these enzymes utilize the N-terminal leader peptide (LP) region of the precursor as a recognition sequence and act on the core peptide (CP) to introduce PTMs such as azole and dehydroalanine (Dha). For pyridine-containing thiopeptides, a pyridine synthase eventually catalyzes formation of a six-membered heterocycle in the CP and eliminates the LP, yielding a macrocyclic thiopeptide. Thus, thiopeptides represent a group of ribosomally synthesized and posttranslationally modified peptide (RiPP) natural products[7].

RiPP biosynthetic logic is highly conducive to bioengineering[8,9]. Simple nucleotide substitutions in the structural gene yield novel compounds, provided that these mutations are tolerated by the biosynthetic machinery. For BGCs encoding promiscuous enzymes, e.g., lanthipeptides and cyanobactins, this strategy can be applied to construct combinatorial libraries of natural product analogs. Recent studies demonstrated that such libraries can be screened to improve or completely reprogram antibacterial activities of the underlying RiPPs[10–17].

In contrast, thiopeptide bioengineering proved to be significantly more challenging. Single-point mutagenesis studies[18–22] and a few complementary reports (e.g., BGC minimization[23] and an incorporation of a single non-proteinogenic amino acid (npAA) suitable for bioconjugation)[24] represent the bulk of the work on this topic. The challenges in thiopeptide bioengineering can be attributed to a highly cooperative, yet only partially understood biosynthesis process. For many thiopeptides, the roles of individual biosynthetic enzymes are only beginning to be elucidated[25–29]. Chemoenzymatic and semisynthetic approaches[30–36] may circumvent the limitations imposed by biosynthetic machinery, but due to the structural complexity of thiopeptides, these strategies present a number of challenges of their own.

We previously reported isolation and characterization of lactazole A, a cryptic thiopeptide from *Streptomyces lactacystinaeus*[37] (Fig. 1a). It is biosynthesized from a compact 9.8 kb *laz* BGC encoding just five enzymes essential for the macrocycle formation (Fig. 1b). Lactazole A has a low Cys/Ser/Thr content, a 32-membered macrocycle, and bears an unmodified amino acid in position 2 (Trp2), all of which are unusual features among thiopeptides[38] (Fig. 1c). Moreover, lactazole A shows no antibacterial activity and its primary biological function remains unknown. Recent bioinformatic studies indicated that the lactazole-like thiopeptides comprise close to half of all predicted thiopeptides (251 out of 508 annotated BGCs) and yet the prototypical *laz* BGC remains the only characterized member of this family to date[39]. Overall, lactazole-like thiopeptides remain a rather enigmatic family of natural products, as close to nothing is known about their function, structural diversity, and biosynthesis.

Intrigued by the uniqueness of *laz* BGC, we set out to reconstitute in-vitro biosynthesis of lactazole A and evaluate its suitability for bioengineering. To this end, we report construction of the FIT-Laz system, a combination of flexible in-vitro translation (FIT) with PTM enzymes from *laz* BGC, as a platform for facile in-vitro synthesis of lactazole-like thiopeptides (Fig. 1d). Taking advantage of the FIT-Laz system, we explore the scope of lactazole biosynthesis and find that *laz* BGC can accommodate substrate variations far beyond other thiopeptide BGCs studied to date. A systematic dissection of the pathway leads to the identification of the minimal lactazole scaffold, a CP with only five amino acids indispensable for the macrocyclization process. Ultimately, we demonstrate that Laz enzymes can accommodate randomization of up to ten consecutive amino acids inside the primary macrocycle, suggesting that the minimal lactazole scaffold is an excellent candidate for bioengineering and may be used to discover artificial thiopeptides with de novo-designed biological activities for drug lead development efforts.

## Results

**In-vitro reconstitution of lactazole biosynthesis**. We began with recombinant production of Laz enzymes in *Escherichia coli* BL21 (DE3). The five enzymes (LazB, LazC, LazD, LazE, and LazF; GenBank accession: AB820694.1, MIBIG accession: BGC0000606) were expressed and purified as soluble His-tagged proteins (Supplementary Fig. 4) and the FIT system was used to establish access to the precursor peptide (LazA; Fig. 2a). Linear DNA template encoding LazA was assembled by PCR and incubated with the in-vitro reconstituted translation machinery from *E. coli* supplemented with T7 RNA polymerase. This scheme for precursor peptide production parallels the previously established FIT-PatD and FIT-GS systems, used for the synthesis of azoline-containing peptides[40] and goadsporin analogs[41], respectively.

With all components in hand, we turned to reconstitution of lactazole biosynthesis. Maturation of goadsporin, a distantly related linear azole-containing RiPP, is initiated with the formation of azoles, whereas Dha installation is dependent on it[42] and biosynthesis of thiomuracin also follows a similar modification order[43]. Based on these results, we hypothesized that azole formation is the starting point in lactazole biosynthesis and therefore attempted to reconstitute the activity of LazDEF (LazD, LazE, and LazF) first. LazDE is a split YcaO cyclodehydratase[38,44] characteristic of thiopeptide BGCs: LazD is predicted to bear a RiPPs recognition element, necessary for LP binding[45], and LazE contains an adenosine triphosphate (ATP)-binding domain[46], utilized for ATP-dependent cyclodehydration of Cys/Ser residues in the CP (Fig. 1d). LazF is an unusual bifunctional protein that features a fusion between a flavin mononucleotide-dependent dehydrogenase, which oxidizes azolines installed by LazDE to azoles[47,48], and a glutamate elimination domain, tentatively participating in the formation of Dha (see below). After LazA precursor peptide expressed with the FIT system (Fig. 2b) was incubated with LazDE, the mixture was treated with iodoacetamide (IAA) and analyzed by liquid chromatography–mass spectrometry (LC-MS). The resulting broad-range extracted ion chromatogram (brEIC; see Methods and Supplementary Figs. 5–8 for detailed description of brEIC) indicated that the LazDE reaction yielded a mixture of two, three, four, and five dehydrations (Fig. 2c). Either LazD or LazE alone had no activity (Supplementary Fig. 9a, b). In contrast, incubating LazA with LazDEF afforded a single product containing four azoles (Fig. 2d). No alkylation occurred on this peptide by IAA, suggesting that all Cys residues were cyclized, and MS/MS analysis of this product supported the native azole pattern, i.e., 3 thiazoles in positions 5, 7, 13, and 1 oxazole in position 11 (Supplementary Fig. 10).

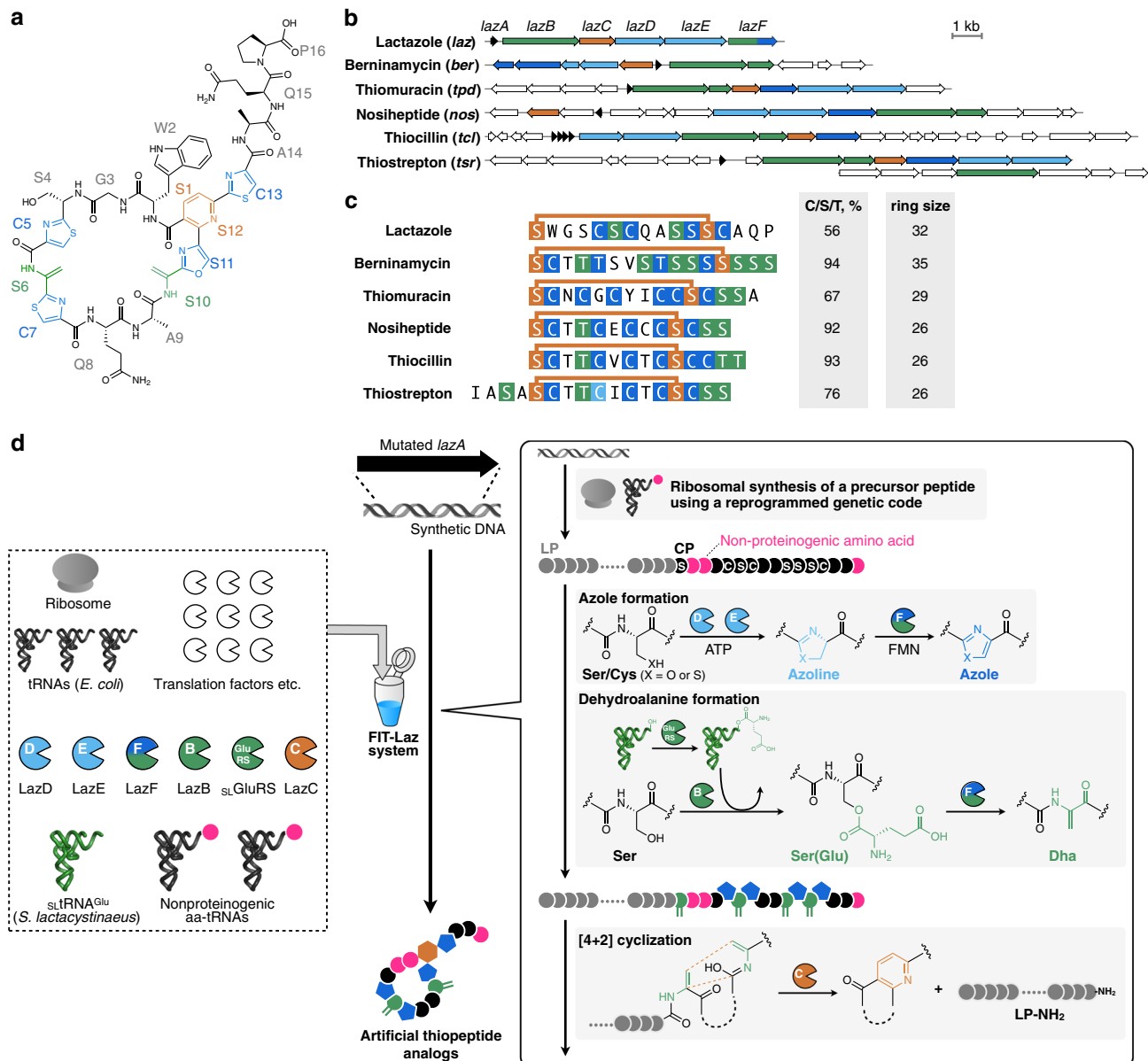

**Fig. 1 Lactazole A and its biosynthesis with the FIT-Laz system. a** Chemical structure of lactazole A. **b** Comparison of *laz* BGC with other prototypical thiopeptide BGCs. Homologs of *laz* genes are color-coded. Genes encoding enzymes responsible for the installation of azolines, azoles, dehydroalanine, and pyridine are shown in light blue, blue, green, and orange, respectively. Precursor peptide structural genes are shown in black and ancillary genes absent from *laz* BGC are in white. **c** Comparison of primary sequences for thiopeptides from **b**, with the same PTM color coding. The comparison reveals an unusual macrocycle size, low C/S/T content, and the absence of azole modification in position 2 as unique features of lactazole. **d** Summary of the FIT-Laz system and the roles of individual enzymes during lactazole biosynthesis. In FIT-Laz, synthetic DNA templates encoding LazA or its mutants are in-vitro transcribed and translated to generate precursor peptides, which undergo a cascade of PTMs introduced by lactazole biosynthetic enzymes to yield lactazole A or its artificial analogs.

Next, we attempted to reconstitute the Dha-forming activity (Fig. 1d). LazBF is a split dehydratase, widely conserved in thiopeptide BGCs[49,50] (Fig. 1b). These proteins are homologous to class I lanthipeptide dehydratases, which utilize Glu-tRNA$^{Glu}$ to glutamylate Ser or Thr residues in the CP of a substrate (using the glutamylation domain)[42], and then catalyze elimination of the glutamate to yield Dha (using the elimination domain). In *laz* BGC, LazB is annotated as a glutamylation domain, and the N-terminal part of LazF is an elimination domain. Even though we assumed that azole formation precedes Dha synthesis, we first attempted to test LazB activity on the unmodified LazA. Surprisingly, LazB glutamylated LazA once when incubated in the presence of synthetic tRNA$^{Glu}$ originating

from *S. lactacystinaeus* and glutamyl-tRNA synthetase (GluRS; GenBank accession: EOY50532.1) from *Streptomyces lividans* (Fig. 2e), whereas reactions lacking any one component led to no modification (Supplementary Fig. 9f–h). These results indicated that similar to homologous enzymes[51,52], LazB utilizes Glu-charged tRNA$^{Glu}$ and catalyzes glutamylation of LazA. As *E. coli* tRNA$^{Glu}$ and GluRS present in the translation mixture were not accepted, LazB appears to be specific for the *Streptomyces* tRNA$^{Glu}$ and GluRS. Complete dehydratase activity was reconstituted with the addition of LazF to the mixture, in which case the reaction yielded a singly dehydrated product (Fig. 2f). Extending the reaction time led to sluggish second and third dehydrations (Supplementary Fig. 9i). These results indicate that

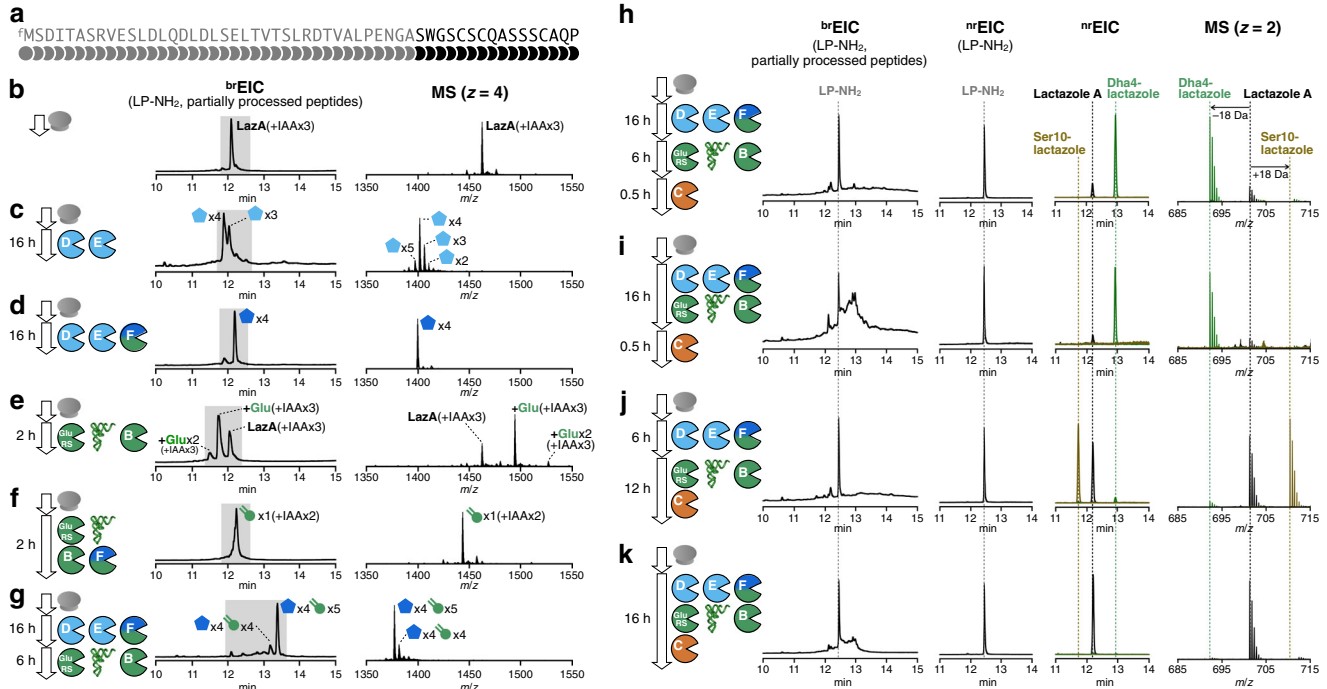

**Fig. 2 Reconstitution of in-vitro lactazole A biosynthesis. a** Primary amino acid sequence of LazA precursor peptide. **b–g** Reconstitution of azole and Dha formation in FIT-Laz. LazA precursor peptide produced with the FIT system was treated with a combination of Laz enzymes as indicated in each panel and the reaction outcomes were analyzed by LC-MS. Displayed are $^{br}$EIC LC-MS chromatograms and composite mass spectra integrated over a time period shaded in the corresponding chromatograms. See Methods and Supplementary Figs. 5–8 for details on reaction conditions and the explanation of $^{br}$EIC chromatograms. **h–k** Reconstitution of lactazole A biosynthesis in FIT-Laz. Displayed are LC-MS chromatograms (left to right: $^{br}$EIC; $^{nr}$EIC at $m/z$ 1026.77 for LP-NH$_2$ generated during the final macrocyclization step; overlaid $^{nr}$EICs at $m/z$ 701.20 for lactazole A shown in black, $m/z$ 692.20 for Dha4-lactazole in green, and $m/z$ 710.20 for Ser10-lactazole in olive) and overlaid mass spectra for lactazole A, Dha4-lactazole, and Ser10-lactazole. These results demonstrate that the order of enzyme addition is critical to the success of lactazole A biosynthesis.

LazBF can catalyze formation of some but not all Dha in LazA independent of azole formation.

We next studied whether LazBF forms the remaining Dha in an azole-dependent manner. To this end, we incubated LazDEF-treated LazA bearing four azoles with LazBF, tRNA$^{Glu}$, and GluRS. This reaction led to a major product 90 Da lighter than the 4-azole LazA, consistent with the formation of five Dha, suggesting that all available Ser in the CP were dehydrated (Fig. 2g). Coincubation of LazA with LazBDEF, tRNA$^{Glu}$, and GluRS resulted in the formation of a complex mixture with the same major product (Supplementary Fig. 9j).

Finally, we tested reconstitution of the entire biosynthetic pathway by adding LazC to the reaction. The putative pyridine synthase LazC has weak sequence similarity to TclM and TbtD, two well-studied enzymes catalyzing analogous reactions[35,53]. Both enzymes are believed to initiate a [4 + 2]-cycloaddition reaction leading to the formation of a macrocyclic product bearing dehydropiperidine, which is further aromatized by eliminating LP as a C-terminal amide (LP-NH$_2$) and a molecule of water to give rise to a pyridine ring. Accordingly, incubation of the aforementioned LazA bearing four azoles/five Dha with LazC afforded LP-NH$_2$ accompanied by a thiopeptide 18 Da lighter than expected, indicating that Ser4, unmodified in lactazole A, was dehydrated (Dha4-lactazole A) (Fig. 2h, i). MS/MS analysis of this product confirmed its structure (Supplementary Fig. 14). Lactazole A was a minor product under these reaction conditions. Changing the order of the enzyme addition (LazDEF followed by LazBC, tRNA$^{Glu}$, and GluRS) suppressed the formation of the overdehydrated product, but instead led to the formation of an underdehydrated thiopeptide, assigned as Ser10-lactazole (Fig. 2j and Supplementary Fig. 15). In contrast to the stepwise reactions,

coincubation of LazA with the full enzyme set (LazBCDEF, tRNA$^{Glu}$, and GluRS) resulted in the formation of lactazole A and LP-NH$_2$, accompanied by minute (<1%) amounts of over- or underdehydrated products (Fig. 2k). LC-MS analysis of the in-vitro-synthesized lactazole A showed that its molecular weight and high performance liquid chromatography (HPLC) retention time were identical to the authentic in vivo-synthetized standard (Supplementary Fig. 12). In addition, both samples had matching, annotatable collision-induced dissociation (CID) MS/MS spectra (Supplementary Figs. 11 and 13), indicating that the one-pot reaction yielded the authentic thiopeptide. Yields of lactazole A and LP-NH$_2$ synthesized in this manner were 18.7 ± 0.4 pmol (±SD from three experiments for 2.5 μl translation reaction scale) for lactazole A and 21 ± 2 pmol for LP-NH$_2$, close to the practical upper limit achievable with reconstituted in-vitro systems (see also Supplementary Methods). These results also suggest that thiopeptide and LP-NH$_2$ formation are coupled to a large degree, and that the thiopeptide production efficiency can be gauged from $^{br}$EIC chromatogram analysis.

In summary, here we demonstrated that the translation product of *lazA* accessed with the FIT system can be treated with the full set of Laz enzymes to yield lactazole A. We refer to this series of transformations as the FIT-Laz system.

**Analysis of substrate tolerance of Laz enzymes.** To understand the overall substrate plasticity of *laz* BGC, we next investigated whether the FIT-Laz system can produce lactazole analogs. We commenced with Ala-scanning mutagenesis and prepared 14 single-point Ala mutants in the CP region of LazA. The precursor peptides were expressed and modified with the FIT-Laz system, and the reaction outcomes were analyzed by LC-MS as above

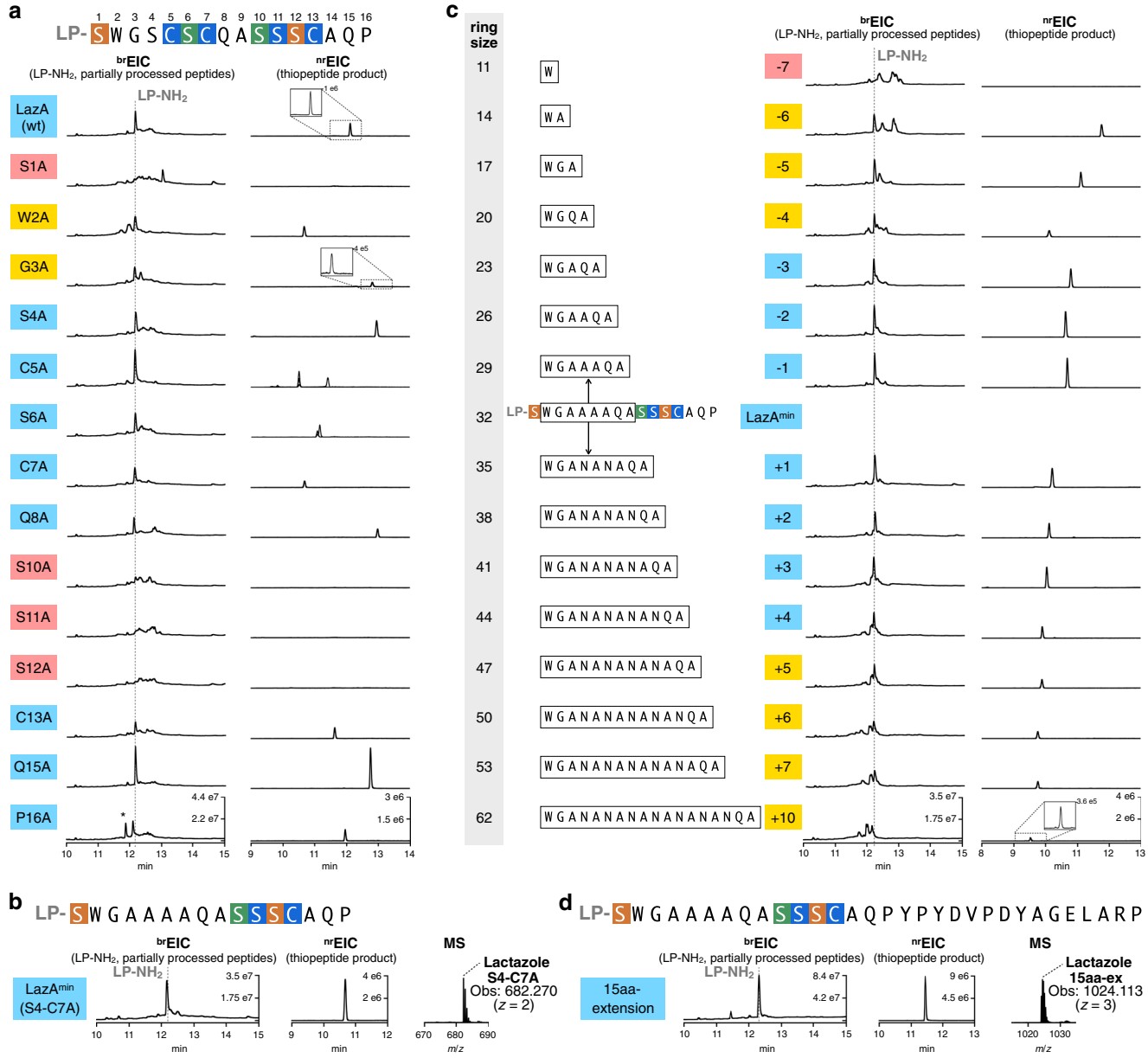

**Fig. 3 Substrate scope of the FIT-Laz system. a** Ala scanning of the LazA CP. Single-point Ala mutants of LazA were treated with the full enzyme set and the outcomes were analyzed by LC-MS. Displayed are LC-MS chromatograms (brEIC chromatograms on the left showing partially processed linear peptides and LP-NH₂ after enzymatic treatment, and nrEIC chromatograms on the right for expected thiopeptides generated at $m/z$ 0.10 tolerance window). For mutants highlighted in light blue biosynthesis proceeded efficiently; yellow highlighting indicates inefficient thiopeptide formation accompanied by the accumulation of linear intermediates and side products; red indicates mutants that failed to yield a detectable thiopeptide. Peaks denoted with an asterisk (*) indicate translation side products. Mutants C5A and S6A gave 4 and 2 thiopeptides, respectively, annotations of which can be found in Supplementary Figs. 16 and 17. Y-axes are scaled between samples for each chromatogram type. **b** LC-MS chromatograms as in **a** for the enzymatic processing of LazA^min on the left with a zoomed-in mass spectrum of the produced thiopeptide on the right. **c** LC-MS chromatograms as in **a** for ring expansion and contraction study of LazA^min. **d** LC-MS chromatograms and mass spectrum as in **b** for a LazA^min variant containing a 15-amino acid extension in the tail region. Collectively, these data point to remarkable substrate tolerance of Laz enzymes.

(Fig. 3a). Only four Ala mutants, S1A, S10A, S11A, and S12A, abolished the formation of thiopeptides, whereas other constructs led to corresponding lactazole analogs and LP-NH₂. During maturation, Ser1 and Ser12 are converted to Dha and are then utilized by LazC for pyridine formation/macrocyclization. Moreover, pyridine synthases are known to recognize the modification pattern around the 4π component[35], which is consistent with the abrogation of biosynthesis in S10A and S11A mutants. On the other hand, C13A mutant was converted to a thiopeptide without modifications in the tail region. Combined with the fact

that the stepwise enzyme treatment led to Ser10-lactazole, these results suggest that the minimal recognition motif around the 4π component for the LazC-catalyzed macrocyclization may be as small as oxazole11-Dha12. Significant accumulation of linear side products and partially processed peptides for W2A and G3A mutants indicates that these amino acids are also important for smooth lactazole biosynthesis. Ala mutants in positions 4–8, 15, and 16, including those disrupting azole and Dha installation, were tolerated, albeit in some cases a mixture of thiopeptides formed (Supplementary Figs. 16 and 17).

Intrigued by these results, we examined whether non-essential modifications inside the macrocycle (Ser4–Cys7) can be removed altogether. Indeed, a tetra-Ala mutant, LazA S4-C7A, was converted to a thiopeptide containing just two azoles and one Dha upon treatment with the full enzyme set (Fig. 3b). A pentamutant LazA S4-C7A, C13A also afforded a thiopeptide, but at a much lower overall efficiency, as a number of partially processed linear peptides accumulated after overnight treatment (Supplementary Fig. 18a). Based on these results, we concluded that the five residues undergoing PTM in LazA S4-C7A (Ser1, Ser10, Ser11, Ser12, and Cys13) are essential for efficient maturation. We termed the resulting thiopeptide as the minimal lactazole scaffold and the corresponding precursor peptide as the minimal lactazole precursor (LazA$^{min}$). As in-vitro biosynthesis of the minimal lactazole proceeded nearly as efficiently as the wild type (Supplementary Fig. 3), we decided to investigate enzymatic processing of LazA$^{min}$ and its potential for bioengineering applications in more detail.

In the next series of experiments, we examined the tolerance of Laz enzymes to the presence of charged amino acids in the CP and performed Lys- and Glu-scanning of LazA$^{min}$ CP. Charged amino acids are rarely found in CPs of thiopeptides[37–39] and RiPP enzymes from other classes are also known to disfavor charged amino acids in general, especially negatively charged Asp and Glu close to the modification site. We prepared 11 single-point Lys mutants and 11 single-point Glu mutants in the non-essential positions of LazA$^{min}$, and analyzed their processing as above (Supplementary Fig. 19). The results of Lys-scanning revealed that a positively charged amino acid is well tolerated in 9 out of 11 positions, whereas W2K and A14K mutants suffered from inefficient processing. Glu was less accepted than Lys overall. In addition to inefficient processing of W2E and A14E, mutants Q8E, A9E, and P16E also resulted in little to no thiopeptide formation. These data suggest that in addition to the five previously identified amino acids, Trp2 and Ala14 also play an important role in LazA$^{min}$ maturation.

Next, we sought to establish the minimal and maximum macrocycle sizes accessible with FIT-Laz. All known thiopeptides range between 26- (thiocillin, thiostrepton, and nosiheptide) and 35-membered macrocycles (berninamycin)[54]. In addition, previously reported bioengineering of thiocillin BGC led to 23-membered artificial variants[55]. Here we prepared amino acid insertion and deletion variants of LazA$^{min}$, and, as before, expressed and modified them with the FIT-Laz system. The results of LC-MS analysis are summarized in Fig. 3c. Deletion of up to 3 amino acids between Ser1 and Ser10 was well tolerated, and led to the efficient formation of 29- to 23-membered thiopeptides. The 4–6 amino acid deletion mutants were also competent substrates and yielded 20- to 14-membered macrocycles, but at relatively low overall efficiencies, as a number of partially processed linear peptides accumulated. Formation of an 11-membered thiopeptide (deletion of 7 residues) was not observed. Thus, it appears that 14-membered thiopeptides are the smallest accessible with Laz enzymes, 9 atoms smaller than the previously smallest thiocillin variants[55]. In contrast, no upper limit on the ring size could be placed. All tested substrates were accepted by the enzymes: the largest synthesized product bore a 62-membered macrocycle, which corresponds to a 10 amino acid insertion. The overall processing efficiency decreased linearly with increasing the cycle size; where LazA$^{min}$ itself was efficiently converted to a macrocycle and LP-NH$_2$, substrates with multiple amino acid insertions had substantial accumulation of linear intermediates and side products. Sequence extension outside of the macrocycle was also easily achievable, as 3 LazA$^{min}$ variants with the C-terminal tail extensions of up to 15 amino acids were efficiently converted to thiopeptides (Fig. 3d and Supplementary Fig. 18b–d).

Encouraged by these results, we examined whether FIT-Laz can accommodate sequence randomization inside the macrocycle. We prepared ten LazA$^{min}$ variants containing ten consecutively randomized amino acids each, which corresponds to the simultaneous insertion of three amino acids and mutation of residues 3–9 in LazA$^{min}$ CP (see Supplementary Methods for sequence choices). Expression and modification of these peptides by FIT-Laz and the subsequent LC-MS analysis (Fig. 4a) revealed that nine out of ten substrates produced thiopeptides as efficiently as LazA$^{min}$. One substrate (10aa-sub-4) led to the formation of multiple thiopeptides owing to three Ser and Thr residues in the inserted region undergoing differential dehydration (Supplementary Fig. 20), and another variant (10aa-sub-10) had a major accumulation of partially processed linear peptides, albeit with detectable formation of the thiopeptide.

Finally, we combined sequence randomization inside the macrocycle with the C-terminal extension and constructed a LazA$^{min}$ mutant with a 34 amino acid-long CP. Despite its size, this substrate efficiently generated a 3.7 kDa thiopeptide when treated with Laz enzymes (Fig. 4b and Supplementary Fig. 21), highlighting the scaffolding ability of key residues in LazA$^{min}$.

Taken together, these data indicate that *laz* BGC is atypically flexible. Many individual enzymes and entire RiPP pathways are similarly promiscuous[11,15], but thiopeptide biosynthesis is usually sensitive to much more modest perturbations. These results point to potential applications of *laz* BGC in bioengineering.

**Synthesis of hybrid thiopeptides with FIT-Laz.** One advantage of the FIT system is its amenability to genetic code reprogramming[56]. Incorporation of multiple npAAs can be achieved by adding appropriate orthogonal tRNAs precharged with npAAs of choice by the use of flexizymes[56] to the translation mixture lacking certain proteinogenic amino acids and cognate aminoacyl-tRNA synthetases. The FIT system was previously used to synthesize peptides containing a variety of npAAs, including D-, β-, N-methylated-, and α,α-disubstituted amino acids, as well as hydroxyacids[57]. Recently, a combination of genetic code reprogramming in the FIT system with a promiscuous RiPP enzyme also enabled synthesis of peptides containing exotic azoline residues[58]. Such npAAs are often found in peptidic natural products, both in RiPPs[7] and in non-ribosomally synthesized peptides (NRPs)[59]. We reasoned that if LazA precursors containing ribosomally installed npAAs are accepted by Laz enzymes, various hybrid thiopeptides may be accessible with the FIT-Laz system.

We began by testing the ability of FIT-Laz to produce N-methylated thiopeptides and prepared 12 *lazA$^{min}$* mutants bearing a single Met codon (AUG) in the CP. The Met codon was reassigned to either N-methylglycine ($^{Me}$Gly) or N-methylalanine ($^{Me}$Ala) by expressing these genes from a Met-depleted translation mixture in the presence of precharged $^{Me}$Gly-tRNA$_{CAU}$ or $^{Me}$Ala-tRNA$_{CAU}$ (see Supplementary Methods for details). Treatment of these translation products with the full enzyme set ($^{Me}$Gly- and $^{Me}$Ala-scanning mutagenesis) and the subsequent LC-MS analysis revealed that, similar to the results of Lys/Glu-scanning, either of the tested N-methylated amino acid was easily accepted in nine positions, whereas mutations at Trp2, Cys13, and Ala14 were detrimental, affording little to no mature thiopeptide (Supplementary Fig. 22).

Next, we tested whether more diverse npAAs can be incorporated into the thiopeptide scaffold following the same logic. For this study, we focused on *lazA$^{min}$* bearing the AUG codon in position 5 and, analogous to the experiments above, prepared LazA$^{min}$ variants containing D-Ala, D-Ser, cycloleucine (cLeu), pentafluorophenylalanine (Phe(F$_5$)), 5-hydroxy-tryptophan (Trp(5-OH)), lactic acid ($^{HO}$Ala), β-Met, and β-homoleucine (β-hLeu). All of these substrates were smoothly converted to the corresponding

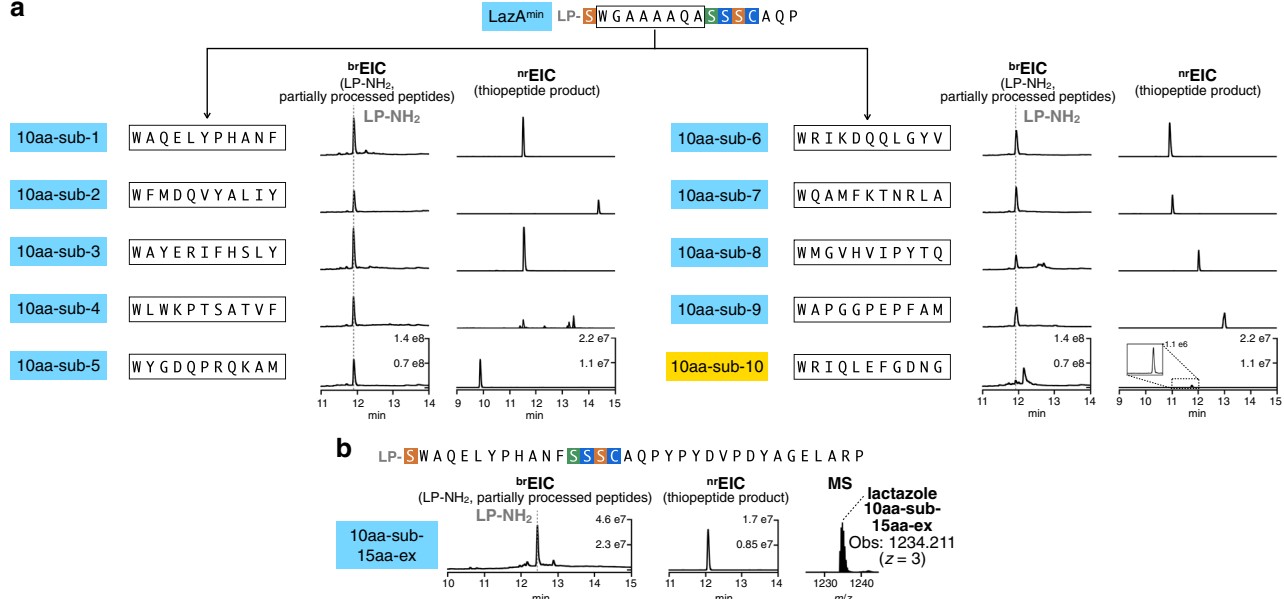

**Fig. 4 Synthesis of lactazole-like thiopeptides with randomized sequences. a** LazA^min variants containing ten consecutively randomized amino acids were first treated with LazDEF and then with LazBC, tRNA^Glu, and GluRS (see Methods for details), and the outcomes were analyzed by LC-MS. Displayed are LC-MS chromatograms (brEIC on the left showing partially processed linear peptides and LP-NH₂ after enzymatic treatment, and nrEIC chromatograms on the right for expected thiopeptides generated at m/z 0.10 tolerance window). Mutants highlighted in light blue indicate efficient thiopeptide assembly; in yellow, inefficient thiopeptide formation accompanied by the accumulation of linear intermediates and side products. One construct, 10aa-sub4, resulted in eight different thiopeptides, partial annotation of which can be found in Supplementary Fig. 20. Efficient in-vitro biosynthesis observed in nine out of ten cases underscores the substrate plasticity of FIT-Laz. **b** LC-MS chromatograms as in **a** for the enzymatic processing of a 34 amino acid-long LazA^min variant on the left with a zoomed-in mass spectrum of the produced thiopeptide on the right.

thiopeptides by the action of Laz enzymes, affording hybrid thiopeptides containing a variety of npAAs (Fig. 5a).

Finally, we studied whether multiple different npAAs can be simultaneously incorporated into the minimal lactazole scaffold to generate highly artificial macrocycles. Due to the presence of a 38-residue LP in LazA^min, the codon boxes available for reprogramming are limited (see Supplementary Methods for details). After some experimentation, we opted to reprogram four codons (AAG, CAU, UGG, and UUU), and reassigned them as ^MeGly, cLeu, Phe(F₅), and ^MeAla, respectively. To this end, a DNA template encoding lazA^min with four codons of interest (Fig. 5b, c) was incubated in a Lys/His/Phe/Trp-depleted translation reaction with ^MeGly-tRNA_CUU, ^MeAla-tRNA_AAA, cLeu-tRNA_GUG, and Phe(F₅)-tRNA_CCA to yield a LazA precursor peptide bearing four npAAs. Treating this substrate with LazBDEF/tRNA^Glu/GluRS afforded a fully processed linear precursor bearing two azoles and three Dha (Supplementary Fig. 23), whereas the reaction utilizing the full enzyme set led to the formation of the predicted thiopeptide accompanied by LP-NH₂ (Fig. 5d). The identities of this macrocycle and its linear precursor were confirmed by CID MS/MS analysis (Supplementary Figs. 23 and 24). From these experiments, we conclude that the FIT-Laz system offers facile access to previously inaccessible hybrid thiopeptides, including heavily modified architectures. These results additionally underscore the promiscuity of Laz enzymes, as all tested substrates containing disruptive amino acids outside of the canonical Ramachandran space were efficiently converted to mature thiopeptides.

## Discussion

In this study, we completed in-vitro reconstitution of laz BGC, which is responsible for biosynthesis of lactazole A, a cryptic thiopeptide from S. lactacystinaeus. The FIT-Laz system

established in this study served as a prototyping tool for rapid probing of lactazole biosynthesis: the entire workflow from PCR assembly of lazA DNA templates to LC-MS analysis of reaction outcomes fits within two working days. An added benefit of working with an in-vitro reconstituted BGC is the ability to decouple self-immunity, export, and proteolytic stability issues, so often complicating in vivo studies, from direct assaying of enzymatic activities. Conversely, in-vitro experiments provide no insight into in vivo fates and metabolism of the underlying natural product, and thus should be interpreted accordingly.

The results presented here indicate that every Laz enzyme tolerates substantial disruptions in the structure of the precursor peptide (Fig. 6a), which stands in contrast to other thiopeptide BGCs characterized to date. Out of 102 structurally diverse precursor peptides tested in this work, 92 yielded lactazole-like thiopeptides, 73 of which were accessed with efficiencies comparable to wild-type lactazole A (Supplementary Data 1). How the enzymes manage to properly modify such a diverse set of substrates remains to be demonstrated. For now, it is apparent that Laz enzymes are highly cooperative. Selective synthesis of the wild-type thiopeptide proceeds only when all enzymes are present in the reaction mixture from the start, whereas any tested stepwise treatments led to either over- or underdehydrated products (Fig. 2h–k). Combined with the fact that some Dha can form independent from azole installation, it is likely that the biosynthetic mechanism is more elaborate than the azoles form first/Dha second model observed during thiomuracin biosynthesis[43,60], and frequently assumed for other RiPPs. Investigations into the nature of this cooperativity are a subject of our ongoing studies.

Broad substrate scope of Laz enzymes enabled development of the minimal lactazole scaffold (Fig. 6a, b). This thiopeptide requires only six PTM events (formation of two azoles, three Dha, and a pyridine heterocycle) for macrocyclization and is

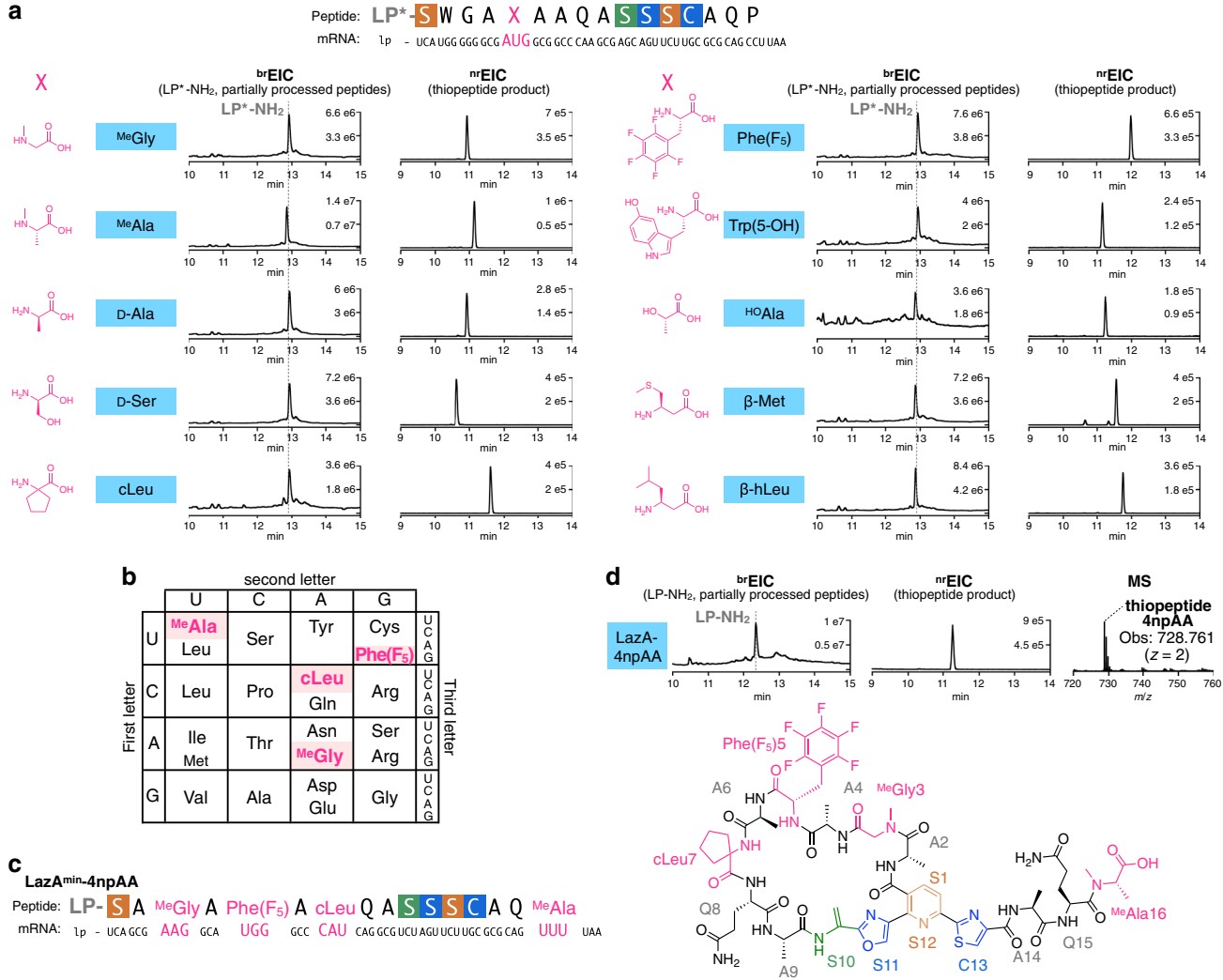

**Fig. 5 Synthesis of hybrid thiopeptides by genetic code reprogramming with the FIT-Laz system. a** Incorporation of a single npAA in a permissible position 5 of LazA$^{min}$ using the Met AUG codon. LazA$^{min}$ mutants accessed with in-vitro genetic code reprogramming were treated with the full enzyme set and the reaction outcomes were analyzed by LC-MS. Displayed are LC-MS chromatograms ($^{br}$EIC chromatograms on the left showing partially processed linear peptides and LP*-NH$_2$ after enzymatic treatment, and $^{nr}$EIC chromatograms on the right for expected thiopeptides generated at $m/z$ 0.10 tolerance window). LP* stands for LazA LP sequence where formyl-Met is replaced with $N$-biotinylated-Phe (see Supplementary Methods for details). **b** Reprogrammed genetic code utilized for the synthesis of an artificial lactazole containing four npAAs and **c** its mRNA sequence. **d** LC-MS chromatograms as in **a** for the enzymatic processing of the LazA$^{min}$ variant from **c**, and the chemical structure of the resulting thiopeptide. Taken together, these data suggest that diverse hybrid thiopeptides are accessible with the FIT-Laz system.

biosynthetically the simplest known thiopeptide to date. The five amino acids undergoing these modifications—Ser1, Ser10, Ser11, Ser12, and Cys13—are indispensable for efficient macrocycle assembly and further experiments demonstrated that the residues adjacent to the modification sites, Trp2 and Ala14, are also important for efficient biosynthesis. Remaining positions (3–9, 15, 16) accept a variety of amino acids, including disruptive npAAs. Modification of minimal lactazole precursor, LazA$^{min}$, in the FIT-Laz system is robust and tolerates massive sequence variations. Specifically, the macrocycle can be contracted or expanded to synthesize 14- to 62-membered thiopeptides (2–18 unmodified amino acids inside the macrocycle; Fig. 6c, d and Supplementary Figs. 25 and 26) and the variants with up to 18 amino acid-long tails are accessible as well (Fig. 6e). Most importantly, LazA$^{min}$ can accommodate mutations of consecutive amino acids, as demonstrated by the synthesis of thiopeptides with ten randomized amino acids inside the macrocycle (Fig. 6e).

The flexibility of the lactazole biosynthetic machinery, combined with the minimal size of *laz* BGC, which contains only the genes essential for macrocyclization, suggest that the minimal thiopeptide scaffold may be an excellent candidate for bioengineering. As continuous randomized sequences can be displayed inside the thiopeptide backbone, we envision that combinatorial libraries based on this scaffold can be generated and screened akin to the recent reports on lanthipeptide bioengineering[10–14]. In those cases, lanthipeptide libraries were prepared with the use of promiscuous lanthipeptide synthases and could be screened against a protein target of interest with the use of phage/yeast display or with the reverse two-hybrid system. These studies resulted in the discovery of lanthipeptide inhibitors of HIV budding process[13], urokinase plasminogen activator[14], and α$_v$β$_3$ integrin binders[11]. Similarly, we anticipate that the integration of the FIT-Laz system with powerful in-vitro screening techniques such as mRNA display[61] will lead to the discovery of artificial

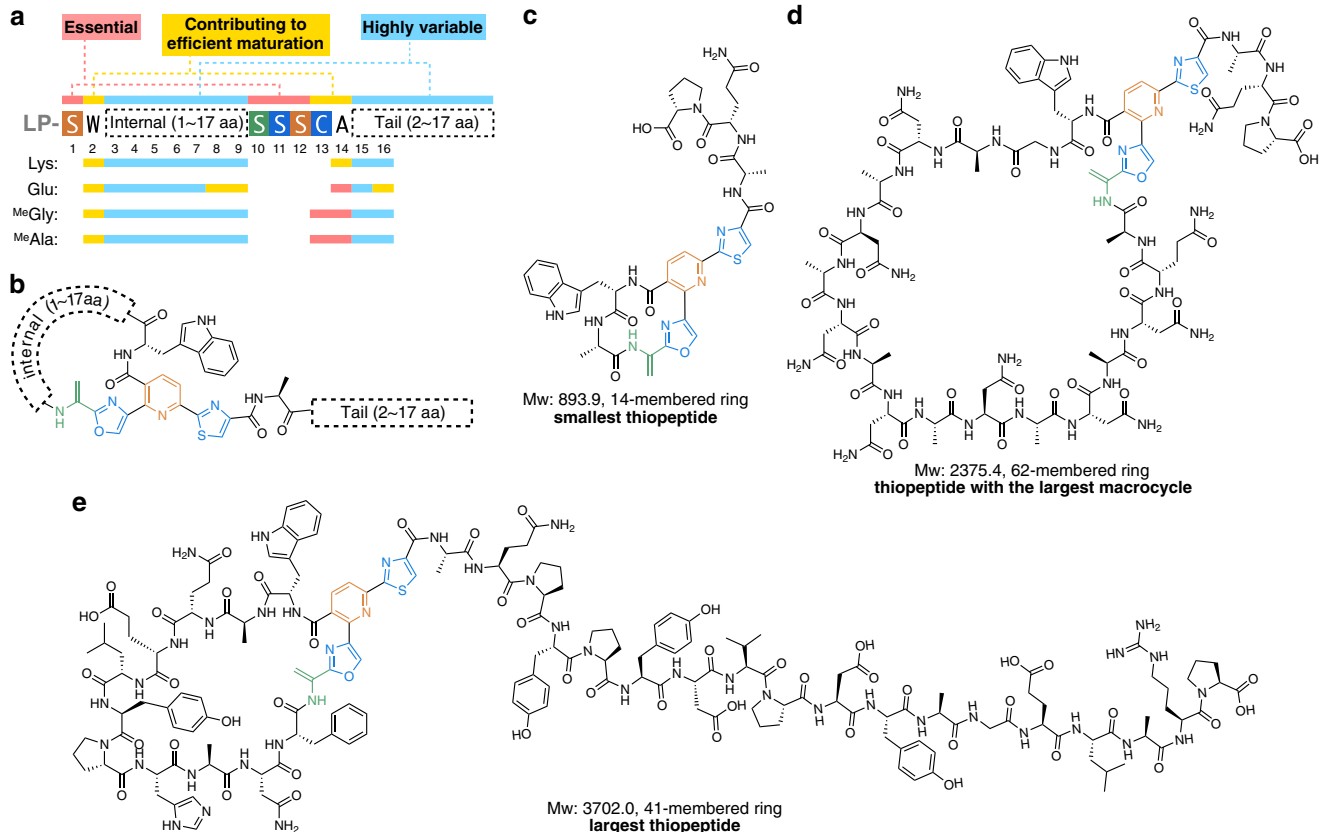

**Fig. 6 Summary of the work. a** Primary sequence representation of the minimal lactazole scaffold with the outcomes of the Lys, Glu, <sup>Me</sup>Gly, and <sup>Me</sup>Ala-scanning experiments mapped to the resulting consensus sequence. **b** Chemical structure representation of the minimal lactazole scaffold. **c–e** Structural diversity of thiopeptides accessible with the FIT-Laz system. Displayed are chemical structures of the smallest artificial lactazole **c**, thiopeptide with the largest macrocycle **d**, and the largest construct **e** synthesized in this work.

thiopeptides with de novo-designed biological activities. Compounds based on the minimal lactazole scaffold may open access to an unexplored chemical space of thiopeptides with desirable pharmacological profiles for drug discovery purposes.

Synthesis of thiopeptide hybrids with other RiPP and NRP classes represents another bioengineering avenue explored in this work. Combinatorial biosynthesis is a concept from NRP and PKS fields, where enzymes from different BGCs are combined to act on a single substrate to generate natural product analogs[62,63]. This concept has recently been applied to RiPPs either via simultaneous use of enzymes from near-cognate BGCs[64] or by devising chimeric LPs[65], demonstrating that multiple promiscuous enzymes can act together to produce nonnatural hybrid RiPPs. In-vitro genetic code reprogramming, easily achievable with FIT-Laz, offers an alternative route to similar hybrids, many of which are inaccessible by existing methods. We demonstrated that thiopeptide-NRP hybrids (macrocycles containing hydroxyacids, D-, β-, N-methylated-, and α,α-disubstituted amino acids), thiopeptide-RiPP hybrids (N-methylation and D-amino acids are found in borosins[66], lanthipeptides[49], proteusins[67], phallotoxins[68], and many other RiPPs families[7]), and thiopeptides with unnatural amino acids (Phe(F₅) and cLeu) can be routinely accessed with FIT-Laz. Such noncanonical hybrid architectures can further expand the range of available molecular complexity for biotechnology and drug discovery. Overall, we believe that the established FIT-Laz system opens exciting opportunities for thiopeptide engineering and characterization of natural thiopeptide diversity.

## Methods

### Gene cloning
Primers used in the gene-cloning experiments and codon-optimized open reading frames (ORFs) are listed in Supplementary Data 2. Genes optimized for recombinant expression in *E. coli* were synthesized by Genewiz in pUC57 vectors with NdeI and XhoI sites flanking each ORF at the 5′- and 3′-ends, respectively. Cloning procedures followed standard molecular biology techniques. For *lazB*, *lazC*, and *lazF*, the codon-optimized sequences (*lazB* and *lazC*) or PCR-amplified ORF (*lazF*; amplified from pKU465-ltc18-6C[37]) were cloned into pET26b (Novagen) using NdeI/XhoI restriction cloning. For *lazD* and *lazE*, the codon-optimized sequences were amplified by PCR using appropriate primers and cloned into pColdII (TAKARA) with NdeI/BamHI restriction enzymes. The identity of all recombinant constructs was assessed by DNA sequencing and correct clones were selected for protein expression.

### Protein expression and purification
For LazD and LazE, *E. coli* BL21(DE3) were transformed with the appropriate pColdII cold-shock plasmids using 100 μg/mL carbenicillin as a selection marker and the resulting transformants were inoculated in Luria-Bertani (LB) medium containing 100 μg/mL carbenicillin. After incubation at 37 °C for 16 h, overnight cultures (20 mL) were inoculated in 800 mL LB with 100 μg/mL carbenicillin, and grown at 37 °C for 3 h. The flasks were then transferred on ice, cooled to 4 °C, and induced with isopropyl

β-D-1-thiogalactopyranoside (IPTG) to a final concentration of 100 μM. Expression was carried out overnight at 15 °C for 20 h, while shaking at 180 r.p.m., and then the cells were collected by centrifugation ($4720 \times g$ for 10 min). The pellets were resuspended in 20 mL lysis buffer (50 mM Tris buffer pH 8.0 containing 500 mM NaCl, 10 mM imidazole, and 2 mM dithiothreitol (DTT)), and lysed by sonication on ice using a UD-100 (TOMY) sonicator. The soluble fraction was separated by centrifugation at 4 °C ($10,300 \times g$ for 30 min), filtered, and loaded on a Bio-Scale TM Mini Profinity TM IMAC cartridge pre-equilibrated with lysis buffer. The column was washed with 50 mL lysis buffer and then eluted with the same buffer containing 200 mM imidazole (pH 8.0). Fractions containing pure protein were combined, desalted with a Bio-Gel P-6 Desalting Cartridge into protein storage buffer (25 mM HEPES pH 8.0, 500 mM NaCl, and 2 mM DTT), and concentrated using a 30 kDa cutoff Amicon centrifugal filter (Merck) to 42 μM and 22 μM for LazD and LazE, respectively. Protein concentrations were determined using 280 nm absorbance using extinction coefficients calculated with the ExPASy ProtParam tool. Purity at various purification stages was assessed by Coomassie-stained SDS-polyactlamide gel electrophoresis analysis. Proteins were flash-frozen and stored at −80 °C.

LazB, LazC, and LazF were expressed in *E. coli* BL21(DE3) transformed with the appropriate pET26b plasmids. Overnight LB cultures (4 mL) were inoculated in 200 mL of ZYM-5052 autoinducing medium[69] supplemented with 100 μg/mL kanamycin and grown at 18 °C for 20 h, while shaking at 180 r.p.m. Purification of these proteins closely followed the protocol described for LazD and LazE, except in these cases lysis buffer lacked DTT and the final storage buffer was different. For LazB, the storage buffer was 25 mM HEPES pH 8.0, 500 mM NaCl, 5% glycerol; for LazC and LazF—25 mM HEPES pH 6.8, 500 mM NaCl. Final protein stock concentrations were 18 μM for LazB, 23 μM for LazC, and 33 μM for LazF. For LazF, the concentration was determined using a Bradford colorimetric assay.

For *S. lividans* GluRS expression, *E. coli* BL21(DE3) transformed with pET26b-*gluRS* were inoculated in LB medium containing 100 μg/mL kanamycin and incubated at 37 °C for 16 h. Then, 4 mL of the overnight culture was inoculated in 200 mL LB medium containing 100 μg/mL kanamycin as a selection marker and the cells were grown at 37 °C for 2 h, while shaking at 150 r.p.m. The flasks were transferred on ice and the culture was induced with IPTG to a final concentration of 100 μM. Expression was done at 18 °C for 20 h while shaking at 180 r.p.m. The purification protocol followed LazD and LazE, except lysis buffer contained 50 mM Tris pH 8.0, 300 mM NaCl, 20 mM imidazole. After His-tag purification, the protein was dialyzed against 50 mM Tris pH 7.5. GluRS was concentrated as described above to a final concentration of 36 μM and flash-frozen for storage.

### Preparation of *lazA* synthetic DNA

Linear double-stranded DNA encoding T7 promoter upstream of *lazA* ORF and mutants of thereof were assembled by PCR from synthetic single-stranded DNA oligonucleotides using Taq polymerase. All PCR were performed in 10 mM Tris pH 8.4, 50 mM KCl, 0.1% (v/v) Triton X-100, 2.5 mM MgCl$_2$, 250 μM each dNTP supplemented with 500 nM of appropriate primers, and Taq DNA polymerase (standard PCR conditions). Three-stage thermal cycling included a denaturing step at 95 °C for 40 s, annealing at 52 °C for 40 s, and extension at 72 °C for 40 s. The list of all oligonucleotides and assembly schemes can be found in Supplementary Data 1.

Briefly, for wild-type *lazA*, forward and reverse primers were annealed and extended in the primer extension step. To this end, 100 μL PCR solution containing 500 nM primers was denatured

at 95 °C for 60 s. Then, five cycles of annealing (52 °C for 60 s) and extension (72 °C for 60 s) were performed, and 1 μL of the product was used as a template for the next step. The first PCR was performed under the standard conditions with five cycles of amplification. The second PCR was performed on a 1000 μL scale using 5 μL of the first PCR product with 14 cycles of amplification, and the outcome was evaluated by agarose gel electrophoresis. DNA was first extracted by phenol/chloroform/isoamyl alcohol (25 : 24 : 1, saturated with 10 mM Tris pH 8.0, 1 mM EDTA) and then by chloroform/isoamyl alcohol (24 : 1). Extracted DNA was precipitated with ethanol, washed with 70% ethanol in water (v/v), and dissolved in 100 μL of water. Other templates were assembled using wild type or *lazA*$^{min}$ as a template in one or two steps. For single-step mutagenesis, reactions were performed on a 200 μL scale with 1 μL of 1 : 100 diluted template DNA and 14 cycles of amplification. For two-step procedures, 100 μL of PCR solution containing 0.5 μL of 1 : 10 diluted template DNA was amplified for 10 cycles, followed by a 14-cycle amplification of the PCR product from the first step (1 μL in 200 μL PCR solution). Analysis of amplification and DNA isolation were performed as above. These DNA templates were used for in-vitro translation as is, without concentration adjustment.

### In-vitro translation and enzymatic reactions

A transcription-coupled in-vitro translation system was reconstituted by mixing purified ribosome, enzymes, and translation factors. The final reaction mixture contained 50 mM HEPES-KOH pH 7.6, 100 mM KOAc, 2 mM guanosine triphosphate, 2 mM ATP, 1 mM cytidine triphosphate, 1 mM uridine triphosphate, 20 mM creatine phosphate, 12 mM Mg(OAc)$_2$, 2 mM spermidine, 2 mM DTT, 1.5 mg/mL *E. coli* total tRNA (Roche), 1.2 μM ribosome, 0.6 μM methionyl-tRNA formyltransferase, 2.7 μM prokaryotic initiation factor-1, 0.4 μM prokaryotic initiation factor-2, 1.5 μM prokaryotic initiation factor-3, 10 μM elongation factor thermo unstable, 10 μM elongation factor thermo stable, 0.26 μM elongation factor G, 0.25 μM release factor-2, 0.17 μM release factor-3, 0.5 μM ribosome recycling factor, 4 μg/mL creatine kinase, 3 μg/mL myokinase, 0.1 μM pyrophosphatase, 0.1 μM nucleotide-diphosphatase kinase, 0.1 μM T7 RNA polymerase, 0.73 μM alanyl-tRNA synthetase (AlaRS), 0.03 μM ArgRS, 0.38 μM AsnRS, 0.13 μM AspRS, 0.02 μM CysRS, 0.06 μM GlnRS, 0.23 μM GluRS, 0.09 μM GlyRS, 0.02 μM HisRS, 0.4 μM IleRS, 0.04 μM LeuRS, 0.11 μM LysRS, 0.03 μM MetRS, 0.68 μM PheRS, 0.16 μM ProRS, 0.04 μM SerRS, 0.09 μM ThrRS, 0.03 μM TrpRS, 0.02 μM TyrRS, 0.02 μM ValRS, 500 μM each proteinogenic amino acid, and 100 μM 10-formyltetrahydrofolate (10-HCO-H4 folate). Translations were performed at 37 °C for 50–60 min with 1 μL of *lazA* variant template DNA, for a total translation volume of 5 μL. For *lazA* variants bearing an amber stop codon (TAG), the reaction mixture was additionally supplemented with 1.00 μM release factor-1.

The translation product was split in two equal parts. One part was treated with 12.5 μL of the Laz enzyme mix, which contained 2 μM LazB, 2 μM LazC, 1 μM LazD, 1 μM LazE, 2 μM LazF, 1 μM *S. lividans* GluRS, 10 μM *S. lactacystinaeus* tRNA$^{Glu}$ in 50 mM Tris buffer pH 8.0 supplemented with 10 mM MgCl$_2$, 5 mM ATP, and 1 mM DTT. The second half was incubated with the same buffer lacking enzymes and tRNA, as a translation control. After a 15–16 h incubation at 25 °C, the reactions were transferred on ice and one volume of 30 mM IAA in methanol was added. Precipitated protein and nucleic acid were separated by centrifugation ($15,300 \times g$ for 4 min), and 10 μL of the supernatant was analyzed by LC-MS. Randomized peptides were modified via a two-step protocol. Translation product (2.5 μL) was first incubated with LazDEF for 6 h and then LazBC, tRNA$^{Glu}$, and GluRS

were added to final concentrations as specified above. After a 12 h incubation, sample preparation and analysis followed the general protocol.

## LC-MS analysis of enzymatic reactions

Waters Xevo G2-XS QTof instrument equipped with Acquity I-Class UPLC system was used for LC-MS analysis. HPLC was done on an Acquity UPLC Peptide BEH C18 column (dimensions: $150 \times 2.1$ mm; pore size: 300 Å; particle size: 1.7 μm) or an identical C4-phase column using 0.1% (v/v) formic acid in water (solvent A) and 0.1% (v/v) formic acid in acetonitrile (solvent B) as a mobile phase. Analysis was performed at 60 °C and 250 μL/min flow rate running the following gradient: 1% B for 2 min; 1–81% B over 20 min; 95% B for 2 min; 1% B for 6 min (total run time: 30 min). MS analysis was done in a positive polarity/high-sensitivity mode with a 0.3 s scan time. Capillary voltage was set to 700 V; electrospray ionization source and desolvation temperatures were 120 °C and 400 °C, respectively. Manufacturer-supplied [Glu1]-fibrinopeptide B was used as a lockspray standard for continuous mass axis referencing and the recommended lockspray setup procedure was performed prior to every run. For MS/MS, a data-dependent acquisition method was used. CID fragmentation was triggered in real time if detected ions met the following conditions: ion intensity exceeded $4 \times 10^4$ ions and the ion charge was equal to 3, 4, or 5 (for analyses involving linear LazA precursors) or $z = 2$ (for thiopeptides). MS/MS spectra were acquired with a 2 s scan time, with parameterized collision energy values. In method 1, collision energies were set to ramp from 6–8 to 30–40 eV over the range of acquired $m/z$ values (200–2000), and in method 2, these values were 6–8 to 45–55 eV. Both methods were utilized for each analyzed peptide and a more informative spectrum was carried forward with. LC-MS data were analyzed with MassLynx v.4.1 and MS/MS assignments were done in Unifi v.1.8.2.169, allowing relevant PTMs where needed. Fragmentation assignments for thiopeptides were done manually.

## LC-MS data analysis

Broad-extracted ion current (broad EIC, $^{br}$EIC) chromatograms were found to be the most informative way to visualize reaction progress (Supplementary Figs. 5–8). Despite methanol precipitation of translation proteins, nucleic acids, and Laz enzymes, significant interference in total ion current (TIC) chromatograms was observed but, in general, most interfering compounds had low molecular weight (<2000 Da) and thus the $m/z$ region above 1000 stayed relatively clean over the course of a run. Linear LazA precursors, modified or not, were predominantly detected as $z = 4$ species, and as such, had $m/z$ values well above 1000. Generating EIC chromatograms for LazA analogs at $z = 4$ with $m/z \pm 100$ (±400 Da) tolerance window enabled visualization of reaction outcomes without interference from the translation components. With the exception of the final macrocyclization reaction, all linear intermediates and side products had mass changes less than the indicated ±400 Da. LP-NH$_2$ also fell within the resulting EIC range at $z = 3$, and thus $^{br}$EIC chromatograms captured all detectable modifications on LazA and were used to judge reaction outcomes in a semi-quantitative manner. For each reaction, the validity of this procedure was cross-checked against manual inspection of a parent TIC chromatogram. The second product of the macrocyclization reaction, a thiopeptide, was outside of this range and had to be visualized separately. For thiopeptides, narrow range EIC ($^{nr}$EIC) chromatograms were generated with an $m/z$ tolerance window generally set to 0.10. When comparing ion intensities of thiopeptides against each other, charge-state series EIC traces were summed to yield a full intensity chromatogram for each thiopeptide. A summary of observed and calculated molecular weights for all discussed compounds can be found in Supplementary Data 1. Details regarding LC-MS quantification of thiopeptide production with FIT-Laz can be found in Supplementary Methods.

**Reporting summary.** Further information on research design is available in the Nature Research Reporting Summary linked to this article.

## Data availability

Primer sequences, codon-optimized ORFs used for protein expression, DNA template assembly schemes, and summary of LC-MS data are available in Supplementary Data 1 and 2. Other results are available upon reasonable request from the corresponding authors.

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

## Acknowledgements

We thank Dr Kazuya Teramoto, Dr Takayuki Kuge, Mr Kakeru Narumi, and Dr Emiko Nagai for their help with protein expression. This work was supported by PRESTO, Japan Science and Technology Agency (JST), to Y.G.; CREST for Molecular Technologies, JST, to H.S.; KAKENHI (JP16H06444 to H.S., Y.G., and H.O.; JP17H04762, JP18H04382, and JP19K22243 to Y.G.) from the Japan Society for the Promotion of Science (JSPS); a grant-in-aid from the Institute for Fermentation, Osaka (IFO), to H.O., S.A., and T.O.; Amano Enzyme, Inc. to H.O. and S.A.; and the A3 Foresight Program, JSPS, to H.O. and S.A.

## Author contributions

Y.G., H.S., and H.O. conceived and supervised the study. A.V., Y.G., T.O., H.S., and H.O. designed experiments. A.V., M.S., Y.G., T.O., S.A. Y.S., and Y.S. performed in-vitro reconstitution experiments. A.V. and M.S. performed substrate tolerance study and A.V. performed genetic code reprogramming experiments. All authors analyzed the experimental results. A.V., M.S., Y.G., S.A., H.S., and H.O. wrote the manuscript with input from all authors. A.V. and Y.G. prepared manuscript figures.

## Competing interests

The authors declare the following competing interests: A.V., Y.G., H.S., and H.O. are listed as co-inventors on a patent application related to this work (PCT/JP2019/38431). All other authors declare no competing interests.
