## [Peer Review File · Nature Communications]

REVIEWERS' COMMENTS:

Reviewer #2 (Remarks to the Author):

In this manuscript, Vinogradov and co-workers describe a remarkably tolerant thiopeptide biosynthetic pathway. All of the major issues raised in an earlier round of review have been corrected and the manuscript is significantly improved as a result. The text is well written and the figures presented in a scholarly fashion. It's an enjoyable paper to read and I recommend publication in Nat Comm, although some minor changes would improve the manuscript further.

Line 45: *Staphylococcus* typo

Page 7-8, line 228-231: This is quite a tenuous claim without any substantive evidence. Handling a large macrocycle is not the same as performing an intermolecular [4+2].

Page 11, line 356: A simple "unnatural amino acid" will do here.

Figure 1: I don't know anyone who calls it a [2+4] cyclization. It's a [4+2] cycloaddition. Also, the authors show that tRNA is needed for Dha formation but they do not show that ATP/FMN are needed for azol(in)e formation. Why show one but not the other?

Reviewer #3 (Remarks to the Author):

This is a tremendous body of work by Suga, Onaka, and coworkers. The technical aspect of the work top notch and will serve as the standard to which RiPP biosynthesis studies will be judged against.

The biggest negative aspect of the work continues to the lack of bioactivity of the lactazole molecule. The authors have done a commendable job of bringing this out in the introduction and the lack of biological testing of the products reported in this study cannot be held against the authors.

A major improvement in this work, as compared to authors' last submission has been the yield measurements. The reviewer recognizes that this is a good starting point and this system can be further improved upon.

All in all, tremendous work and extremely well polished presentation. The reviewer has no concerns in recommending publication.

This reviewer also commented on whether reviewer #1's concerns have been addressed:

Reviewer 1 expressed two major reservations, first being the lack of yield measurements for the FIT-Laz system. This concern has been experimentally addressed by the authors now.

Quantification of yields is now established.

The second concern was the lack of biological activity, and this concern was raised by other reviewers also. The reviewer had commented that library of lactazole derivatives could be made using the FIT-Laz system and that this should be formally explored here. I agree with the authors' answer to this concern that this is beyond the scope of this work. What is being shown here is the applicability of the FIT-Laz system to demonstrate the promiscuity of the RiPP biosynthetic system. Future work will undoubtedly involve optimizations on this system to generate libraries. The

experimental work and findings presented in this manuscript, in its present form, justifies its motivations.

All in all, I would say that the concerns expressed by Reviewer#1 have been adequately addressed by the authors.

Reviewer #1 (Remarks to the Author):

While the authors have now included yields for their synthesis which has addressed a reviewer concern, this paper still falls short. As it stands, the paper is an investigation of the (surprising) promiscuity of the lactazole A biosynthetic machinery. This is a decent use of the FIT system, but the paper is still thin on impact. Doing an in vitro selection using their new machinery would bring it up to the next level. Another downstream issue is that any analogs they do find will be hampered by challenges in scale-up which requires significant effort. This effort would be justifiable if they had identified a potent hit. Without this, it is just an enzymatic pathway substrate tolerance story, worth publishing, but in a lower-tier journal.

1 Response to the reviewers of the manuscript “Minimal lactazole
2 scaffold for in vitro thiopeptide bioengineering”

3 Reviewer #2

4 > In this manuscript, Vinogradov and co-workers describe a remarkably tolerant
5 thiopeptide biosynthetic pathway. All of the major issues raised in an earlier round of
6 review have been corrected and the manuscript is significantly improved as a result. The
7 text is well written and the figures presented in a scholarly fashion. It's an enjoyable paper
8 to read and I recommend publication in Nat Comm, although some minor changes would
9 improve the manuscript further

10 Thank you so much for taking your time to thoroughly evaluate the manuscript! Please find
11 our responses to your suggestions below.

12 > Line 45: *Staphylococcus* typo

13 Lines 43-44 now read: “including methicillin resistant *Staphylococcus aureus* (MRSA).”

14 > Page 7-8, line 228-231: This is quite a tenuous claim without any substantive evidence.
15 Handling a large macrocycle is not the same as performing an intermolecular [4+2].

16 Comparison of LazC to TbtD has been removed altogether.

17 > Page 11, line 356: A simple "unnatural amino acid" will do here.

18 Lines 364-365 now read: “thiopeptides with unnatural amino acids (Phe(F5) and cLeu)
19 <...>”

20 > Figure 1: I don't know anyone who calls it a [2+4] cyclization. It's a [4+2] cycloaddition.
21 Also, the authors show that tRNA is needed for Dha formation but they do not show that
22 ATP/FMN are needed for azol(in)e formation. Why show one but not the other?

23 Figure 1 has been revised to address both suggestions.

24

25 **Reviewer #3**

26 > This is a tremendous body of work by Suga, Onaka, and coworkers. The technical
27 aspect of the work top notch and will serve as the standard to which RiPP biosynthesis
28 studies will be judged against.

29 The biggest negative aspect of the work continues to the lack of bioactivity of the lactazole
30 molecule. The authors have done a commendable job of bringing this out in the
31 introduction and the lack of biological testing of the products reported in this study cannot
32 be held against the authors.

33 A major improvement in this work, as compared to authors' last submission has been the
34 yield measurements. The reviewer recognizes that this is a good starting point and this
35 system can be further improved upon.

36 All in all, tremendous work and extremely well polished presentation. The reviewer has no
37 concerns in recommending publication.

38 This reviewer also commented on whether reviewer #1's concerns have been addressed:

39 Reviewer 1 expressed two major reservations, first being the lack of yield measurements
40 for the FIT-Laz system. This concern has been experimentally addressed by the authors
41 now. Quantification of yields is now established.

42 The second concern was the lack of biological activity, and this concern was raised by
43 other reviewers also. The reviewer had commented that library of lactazole derivatives
44 could be made using the FIT-Laz system and that this should be formally explored here. I
45 agree with the authors' answer to this concern that this is beyond the scope of this work.
46 What is being shown here is the applicability of the FIT-Laz system to demonstrate the
47 promiscuity of the RiPP biosynthetic system. Future work will undoubtedly involve
48 optimizations on this system to generate libraries. The experimental work and findings
49 presented in this manuscript, in its present form, justifies its motivations.

50 All in all, I would say that the concerns expressed by Reviewer#1 have been adequately
51 addressed by the authors.

52 Thank you for the warm appraisal of our work!

53

54 **Reviewer #1**

55 > While the authors have now included yields for their synthesis which has addressed a
56 reviewer concern, this paper still falls short. As it stands, the paper is an investigation of
57 the (surprising) promiscuity of the lactazole A biosynthetic machinery. This is a decent use
58 of the FIT system, but the paper is still thin on impact. Doing an in vitro selection using
59 their new machinery would bring it up to the next level. Another downstream issue is that
60 any analogs they do find will be hampered by challenges in scale-up which requires
61 significant effort. This effort would be justifiable if they had identified a potent hit. Without
62 this, it is just an enzymatic pathway substrate tolerance story, worth publishing, but in a
63 lower-tier journal.

64 Thank you for providing valuable criticism and raising a number of critical concerns during
65 the first round of review. We believe that our revised work will be sufficiently interesting to
66 the diverse readership of Nature Communications, an opinion strongly shared by two other
67 reviewers. Importantly, our results lay down foundation for rapid (RaPID) screening for
68 bioactive lactazoles, and we believe that this reviewer will recognize the significance of our
69 work as the foundation for next-generation bioengineering of thiopeptides.